# Inhibition by small-molecule ligands of formation of amyloid fibrils of an immunoglobulin light chain variable domain

**Boris Brumshtein[1,2,3]\*[†], Shannon R Esswein[1,2,3]\*[†], Lukasz Salwinski[1,2,3], Martin L Phillips[2,3], Alan T Ly[1,2,3], Duilio Cascio[1,2,3], Michael R Sawaya[1,2,3], David S Eisenberg[1,2,3]\***

[1]Department of Biological Chemistry, Howard Hughes Medical Institute, UCLA, Los Angeles, United States; [2]UCLA-DOE Institute for Genomics and Proteomics, Los Angeles, United States; [3]Department of Chemistry and Biochemistry, UCLA, Los Angeles, United States

**Abstract** Overproduction of immunoglobulin light chains leads to systemic amyloidosis, a lethal disease characterized by the formation of amyloid fibrils in patients' tissues. Excess light chains are in equilibrium between dimers and less stable monomers which can undergo irreversible aggregation to the amyloid state. The dimers therefore must disassociate into monomers prior to forming amyloid fibrils. Here we identify ligands that inhibit amyloid formation by stabilizing the Mcg light chain variable domain dimer and shifting the equilibrium away from the amyloid-prone monomer.

**\*For correspondence:** boris@mbi.ucla.edu (BB); sesswein@mbi.ucla.edu (SRE); david@mbi.ucla.edu (DSE)

[†]These authors contributed equally to this work

**Competing interests:** The authors declare that no competing interests exist.

## Introduction

Amyloid fibrils are protein deposits often associated with disease. These deposits show common structural and biochemical characteristics, including non-covalent self-complementary fibrils, resistance to proteolysis, insolubility in aqueous solution, binding of thioflavin dye, and generation of a cross-β X-ray diffraction pattern. The spine of amyloid fibrils, termed a 'steric zipper,' is built by paired β-sheets with interdigitating side chains and serves as the scaffold for fibril extension. The inter-strand and inter-sheet distances give rise to a characteristic cross-β X-ray diffraction pattern of two perpendicular reflections at ~4.8 and ~10 Å (*Geddes et al., 1968*; *Hobbs, 1973*; *Sipe and Cohen, 2000*; *Nelson et al., 2005*). Formation of a steric zipper requires the presence of a discrete peptide segment with an amino acid sequence capable of forming a self-complimentary propagating structure. Such segments are commonly found as integral parts of globular proteins and trigger aggregation into amyloid fibrils only upon exposure to solvent through full or partial denaturation (*Nelson et al., 2005*).

Cases of systemic light chain amyloidosis (AL), associated with multiple myeloma, were first described in the mid-19[th] century; however, the etiology of the disease was not clear. In 1848, Dr. Bence-Jones discovered the disease-associated substance in urine, though it was only later identified as immunoglobulin light chains (LC) associated with amyloidosis (*Kyle, 2001*; *2011*; *Dahlin and Dockerty, 1950*; *Glenner et al., 1970*; *1971*; *Glenner, 1973*; *Sipe et al., 2014*). Systemic AL amyloidosis is characterized by overexpression of monoclonal LCs, which are able to form homo-dimers. Despite renal clearance of excess protein, the full-length LCs or their fragments still form amyloid fibril deposits in patients' tissues, most frequently in the heart and kidneys, which result in organ

**eLife digest** Systemic light chain amyloidosis is a disease that occurs when individuals produce too much of an immune protein. The excess protein chains normally exist in the body as individual molecules called "monomers" or in pairs called "dimers," and they can readily switch between these two forms. However, the monomers are also prone to forming amyloid fibrils, which are difficult to break down. Amyloid fibrils are often deposited in the heart and kidneys and can lead to organ failure and death.

Finding molecules that prevent the formation of amyloid fibrils could help to develop treatments for amyloidosis. Now, Brumshtein, Esswein *et al*. have screened 27 compounds to identify those that stabilize the dimer form of the protein. This would reduce the number of monomers in the body, and so reduce the number of immune proteins that can form amyloid fibrils.

The experiments identified four compounds that could stabilize the dimers, including one called methylene blue. Comparing the chemical structures of these compounds with the structures of drugs approved for medical use identified thirteen drugs. However, follow-up tests showed that only one, called sulfasalazine, reduced the formation of amyloid fibrils. Neither methylene blue nor sulfasalazine is likely to have a strong enough effect to treat amyloidosis, but they may serve as templates for future drug designs.

failure (*Falk et al., 1997*; *Buxbaum, 1986*). Although it is clear this deposition causes organ dysfunction, much remains to be discovered about the molecular process by which immunoglobulins assemble into amyloid fibrils. It is unknown which peptide segments form steric zippers and how to block their assembly into the amyloid spine.

LCs are produced by plasma cells and their final amino acid sequence is determined by somatic recombination (*Sakano et al., 1979*; *Marchalonis and Schluter, 1989*). LCs are classified by their amino acid sequence as either $\kappa$ or $\lambda$. They consist of two domains, the variable ($V_L$) and constant ($C_L$), which are connected by a joining segment (J). In patients, amyloid fibrils are found to include either $V_L$s or full-length LCs ($V_L$-J-$C_L$), yet the ubiquitous presence of $V_L$s indicates that this domain may be the minimal and essential unit for fibril assembly (*Falk et al., 1997*; *Buxbaum, 1986*; *1992*; *Olsen et al., 1998*; *Lavatelli et al., 2008*; *Vrana et al., 2009*; *Bodi et al., 2009*).

The crystal structures of full-length LCs and structures of just their $V_L$s both reveal homo-dimers, which resemble the light chain and heavy chain hetero-dimer of the antigen-binding fragment (Fab) of a full antibody. Fab is a hetero-dimer composed of both light and heavy chains, each with constant ($C_L$ and $C_H$) and variable ($V_L$ and $V_H$) immunoglobulin domains. LC homo-dimers, in which the $V_L$ domain is covalently linked to the $C_L$ domain, resemble the quaternary structure of the Fab hetero-dimer, and non-covalently linked dimers of only $V_L$s structurally resemble the variable region of the Fab ($V_L$ and $V_H$) (*Edmundson et al., 1969*; *Colman et al., 1977*; *Firca et al., 1978*). The interfaces between the variable domains of physiological hetero-dimers and pathological homo-dimers are lined with apolar residues encompassing a hydrophobic cavity capable of accommodating small molecules (*Edmundson et al., 1984*; *1993*).

Our work focuses on a $\lambda$ patient-derived $V_L$ homo-dimer named Mcg whose structure and properties were detailed in pioneering studies by Edmundson and coworkers (*Edmundson et al., 1969*; *1984*; *1993*; *Firca et al., 1978*). The cavity between the two $V_L$ domains in Mcg is capable of binding a wide array of ligands, including synthetic organic molecules and peptides, which form contacts with the side chains of the dimer cavity. The cavity spans most of the 24 Å-long dimer interface; it is cylindrically shaped and can be considered to constitute three sites: A, B and C (*Edmundson et al., 1984*). Based on the geometry of the dimer with some deviation from the original nomenclature by Edmundson *et al.*, we designate side chains of residues Y34, Y93, D97 and F99 on one side of the cylindrical cavity as the 'A-site,' S36, Y51, E52, S91 and F101 as the 'B-site,' and Y38, Q40, V48 and Y89 as the 'C-site' (*Figure 1*). Some ligands bind the cavity in a sequential manner without a distinct preference for one of the sites. For example, 1-anilinonaphthalene-8-sulfonic acid binds in either the A or C-sites and can migrate between them, whereas fluorescein binds to both sites with partial occupancy. Other molecules, such as menadione, preferentially bind to only one of the sites

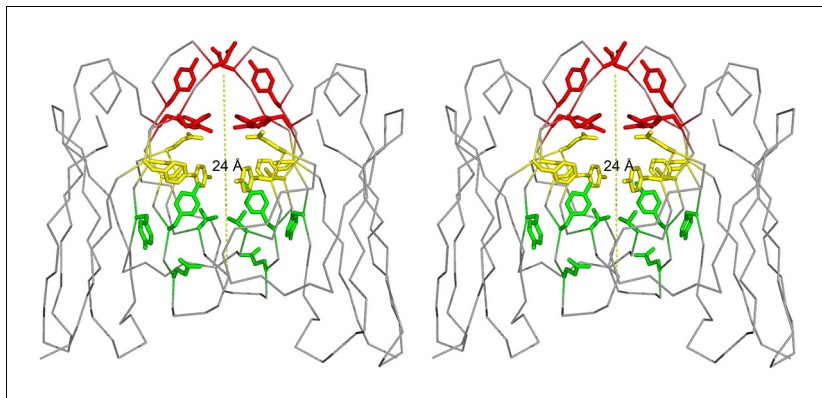

**Figure 1.** Stereo image of the ligand-binding sites of the $V_L$ dimer. We designate the A-site in red (residues Y34, Y93, D97 and F99), B-site in yellow (residues S36, Y51, E52, S91 and F101), and C-site in green (residues Y38, Q40, V48 and Y89).

(*Edmundson et al., 1984*). Regardless of binding location, this demonstrates the dimer cavity is capable of accommodating various hydrophobic and aromatic ligands.

$V_L$s exist in equilibrium between homo-dimers and amyloid-prone monomers. Experiments conducted in denaturing conditions indicate that reducing the stability of the monomeric state promotes amyloid fibril formation, and mutations that induce dimer disassociation or promote monomer unfolding increase the propensity to form amyloid fibrils (*Bernier and Putnam, 1963*; *Kishida et al., 1975*; *Qin et al., 2007*; *Wetzel, 1994*; *Hurle et al., 1994*; *Brumshtein et al., 2014*; *Baden et al., 2008*). Likewise, mutations that stabilize the structure of $V_L$s or covalently fix $V_L$ dimers inhibit formation of amyloid fibrils. These results indicate that formation of amyloid fibrils involves two steps: $V_L$ dimer disassociation into monomers followed by partial or full unfolding. The mechanism of amyloid formation also suggests that shifting the equilibrium away from the amyloid-prone monomer by stabilizing the dimer would hinder formation of amyloid fibrils (*Figure 2*) (*Bulawa et al., 2012*; *Bellotti et al., 2000*).

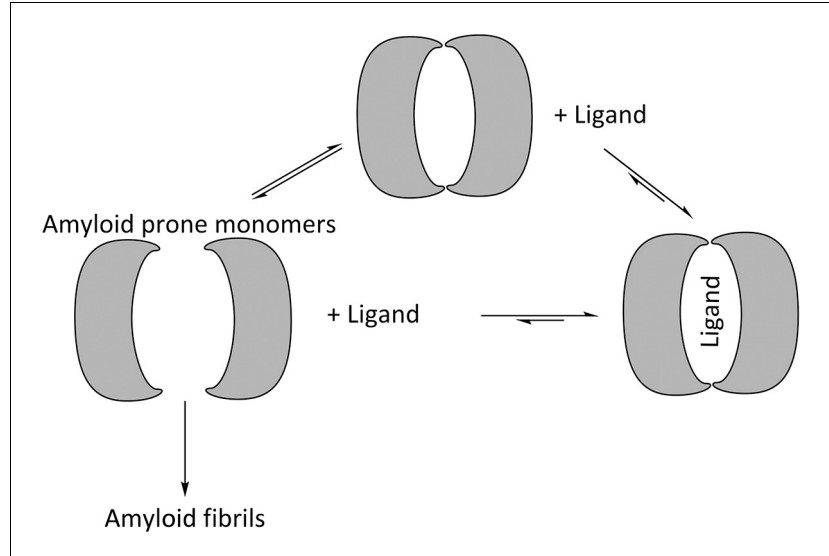

**Figure 2.** Proposed mechanism for using ligands to hinder the aggregation of immunoglobulin $V_L$ s into amyloid fibrils. $V_L$ s are in equilibrium between dimers and monomers in solution. Ligands may be used to stabilize the $V_L$ dimer and therefore shift the equilibrium away from amyloid-prone monomers.

The monomer-dimer equilibrium of $V_L$s suggests that systemic AL amyloidosis may be mitigated by binding ligands to the cavity at the $V_L$ dimer interface (*Figure 2*). This approach proved effective for transthyretin-related amyloidosis, another type of systemic amyloidosis for which stabilizing the quaternary state led to the development of therapeutics (*Miroy et al., 1996*). Upon transthyretin tetramer disassociation into amyloid-prone monomers, it forms amyloid fibrils in an acidic environment. The binding of thyroxine inhibits disassociation and subsequent amyloid formation (*Baures et al., 1998*). Following the same principle, a modified ligand with a disassociation constant in the nanomolar range prevents transthyretin from forming amyloid fibrils and is effective in vivo.

Here we apply structural and biochemical methods to investigate ligands that hinder amyloid formation by stabilizing the $V_L$ homo-dimer. We identify ligands that may serve as prototypes for therapies for treating LC amyloidosis and our results are consistent with a mechanism for amyloidosis that proceeds via dimer disassociation to amyloid-prone monomers (*Qin et al., 2007*; *Brumshtein et al., 2014*).

## Results

*Identification of ligands that inhibit formation of amyloid fibrils.* Based on the previous work of Edmundson *et al.*, we screened a panel of 27 hydrophobic and aromatic ligands for an inhibitory effect on the formation of amyloid fibrils (*Figure 3*) (*Edmundson et al., 1976*; *1984*; *1993*). Such ligands may stabilize $V_L$ dimers and decrease disassociation into amyloid-prone monomers. Varying concentrations of ligands up to 1 mM were used to evaluate the effect of complexes on formation of amyloid fibrils. Formation of amyloid fibrils was monitored using both ThT fluorescence assays and EM micrographs. Molecules that showed an inhibitory effect with concentrations under 1 mM were 8-anilino-1-naphthalene sulfonic acid [5], methylene blue [10], Chicago Sky Blue 6B [14] and Phenol Red [20] (*Figure 3B*).

Methylene blue, Chicago Sky Blue 6B, and Phenol Red (10, 14 and 20) were used to search Drug-Bank for approved biomedical compounds resembling their chemical structures and properties with a similarity threshold value of 0.3 (*Law et al., 2014*). The search identified 13 compounds that were tested for an amyloid inhibitory effect by means of ThT assays and EM analysis (*Figure 3C*). Of these 13 molecules from DrugBank, only sulfasalazine [34] showed an inhibitory effect on formation of amyloid fibrils. Among all the molecules tested, including the ligands initially screened and the subsequent approved biomedical compounds, the two that show the most potent effect are methylene blue [10] and sulfasalazine [34]; both hinder amyloid fibril formation at effective concentrations of 0.1 and 0.5 mM respectively (*Figure 4*).

*Equilibrium dialysis binding of methylene blue and sulfasalazine.* Equilibrium dialysis was used to assess the binding constants of methylene blue and sulfasalazine to Mcg. Measured concentrations were fit to the corresponding model equations and their curves were represented as binding and Scatchard plots (*Figure 5*) (*Scatchard, 1949*; *Spitzer and McDonald, 1956*). The constants were derived from a least squares fit of equations to data and are given in *Table 1*. Although both methylene blue and sulfasalazine bind to Mcg, the Scatchard plots indicate that binding proceeds through somewhat different pathways: methylene blue shows positive cooperative binding, signifying at least two sites with different binding constants, while sulfasalazine shows no cooperativity and suggests an additional, non-specific binding site (*Figure 5*). The best fit for the sulfasalazine-binding data was achieved using a model for two identical, independent binding sites per $V_L$ dimer, followed by non-specific binding.

*Crystal structures of methylene blue and sulfasalazine bound to the $V_L$ dimer.* In the crystal structures of Mcg with methylene blue and sulfasalazine, the ligands bind at the cavity between the two $V_L$ domains (*Figure 6*, *Table 2*). In the structure of Mcg with methylene blue, one ligand is bound to the A-site of the dimer. This differs from equilibrium dialysis results in solution, which indicate at least two methylene blue binding sites. The structure of Mcg with sulfasalazine indicates two symmetry-related molecules bound in the hydrophobic cavity of the dimer with both ligands simultaneously binding to three sites of the $V_L$ dimer cavity.

*Effect of ligand binding on quaternary state.* We used analytical ultracentrifugation to assess whether ligand binding shifts the equilibrium of Mcg towards more dimers or whether ligands bind to monomers as well. The wavelengths used for optical density readings of methylene blue (550 nm) and sulfasalazine (420 nm) did not overlap with protein absorbance. Distributions of molecular

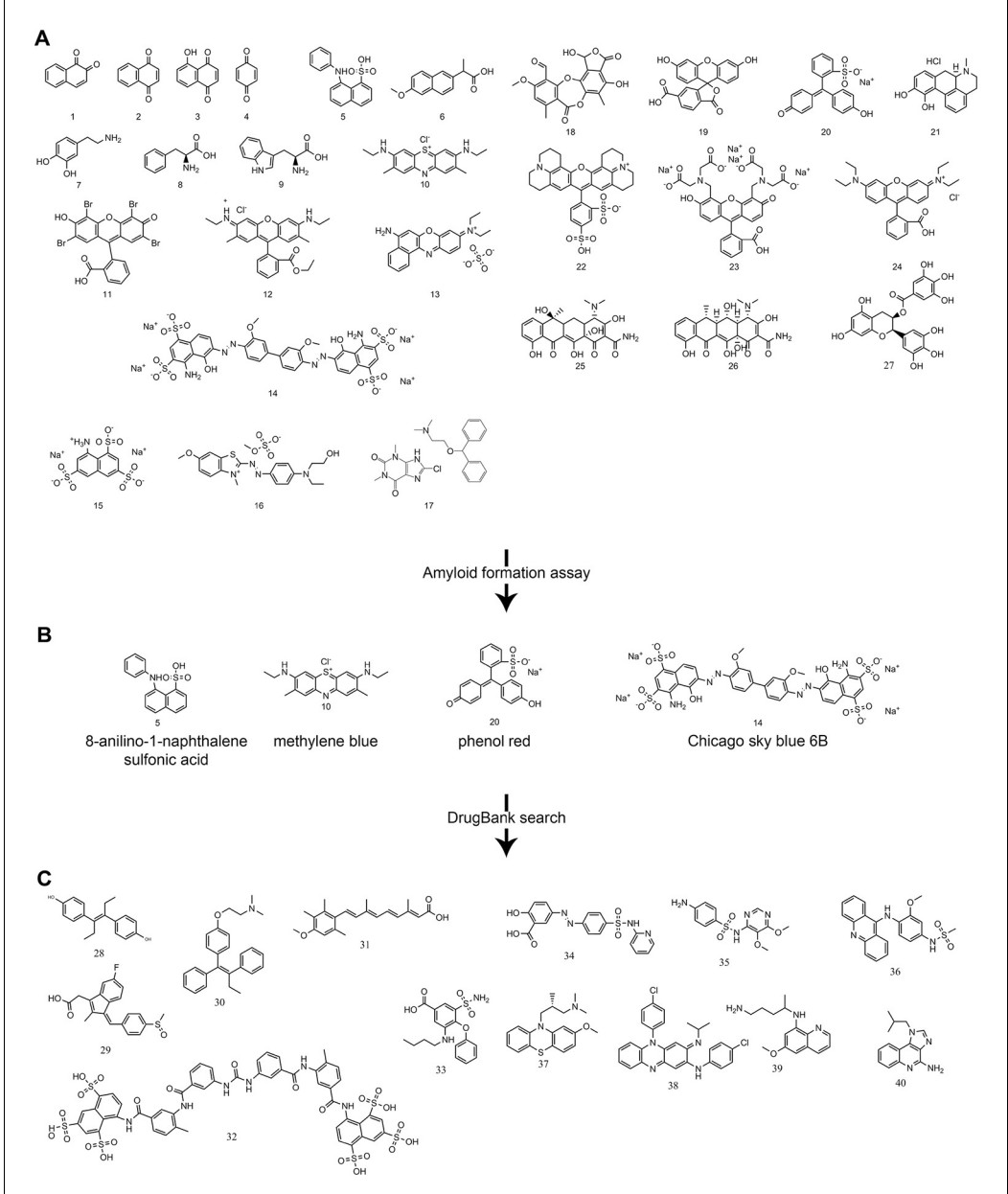

**Figure 3.** Panel of ligands analyzed with ThT assays and EM for their ability to inhibit $V_L$ amyloid formation. (**A**) Preliminary molecules screened. 1. 1,2-naphthoquinone. 2. 1,4-naphthoquinone. 3. 5-hydroxy-1,4-naphthoquinone. 4. quinone. 5. 8-anilinonaphthalene-1-sulfonic acid. 6. methoxy-2-naphthyl-propionic acid. 7. dopamine. 8. L-phenylalanine. 9. L-tryptophan. 10. methylene blue. 11. eosin Y. 12. rhodamine 6G. 13. Basic Blue 12. 14. Chicago Sky Blue 6B. 15. 8-aminonaphthalene-1,3,6-trisulfonic acid. 16. Basic Blue 41. 17. dimenhydrinate. 18. stictic acid. 19. 6-carboxyfluorescein. 20. phenol red. 21. R-(-)-apomorphine. 22. sulforhodamine 101. 23. fluorescein methyleneiminodiacetic acid. 24. rhodamine B. 25. tetracycline. 26. doxycycline. 27. (—)-epigallocatechin gallate. (**B**) Molecules showing an inhibitory effect on the formation of amyloid fibrils during the preliminary screen. (**C**) Molecules identified in a DrugBank search using the most effective candidates from the preliminary screen and a similarity threshold of 0.3. Search using phenol red: 28. diethylstilbestrol (0.4), 29. sulindac (0.4), 30. tamoxifen (0.4), 31. acitretin (0.4). Search using Chicago Sky Blue 6B: 32. suramin (0.4), 33. Bumetanide (0.3), 34. sulfasalazine (0.3), 35. sulfadoxine (0.3), 36. amsacrine (0.3), 37. methotrimeprazine (0.3). Search using methylene blue: 38. clofazimine (0.3), 39. primaquine (0.3), 40. imiquimod (0.3).

weights were fit to the absorbance data, thus permitting detection of only the apparent molecular weights of the protein in complex with ligand (**Figure 7**, **Table 3**). The observed molecular weights indicate that ligands bind only to $V_L$ dimers. For methylene blue, the measured molecular weight was slightly larger than expected; therefore, we performed an additional ultracentrifugation

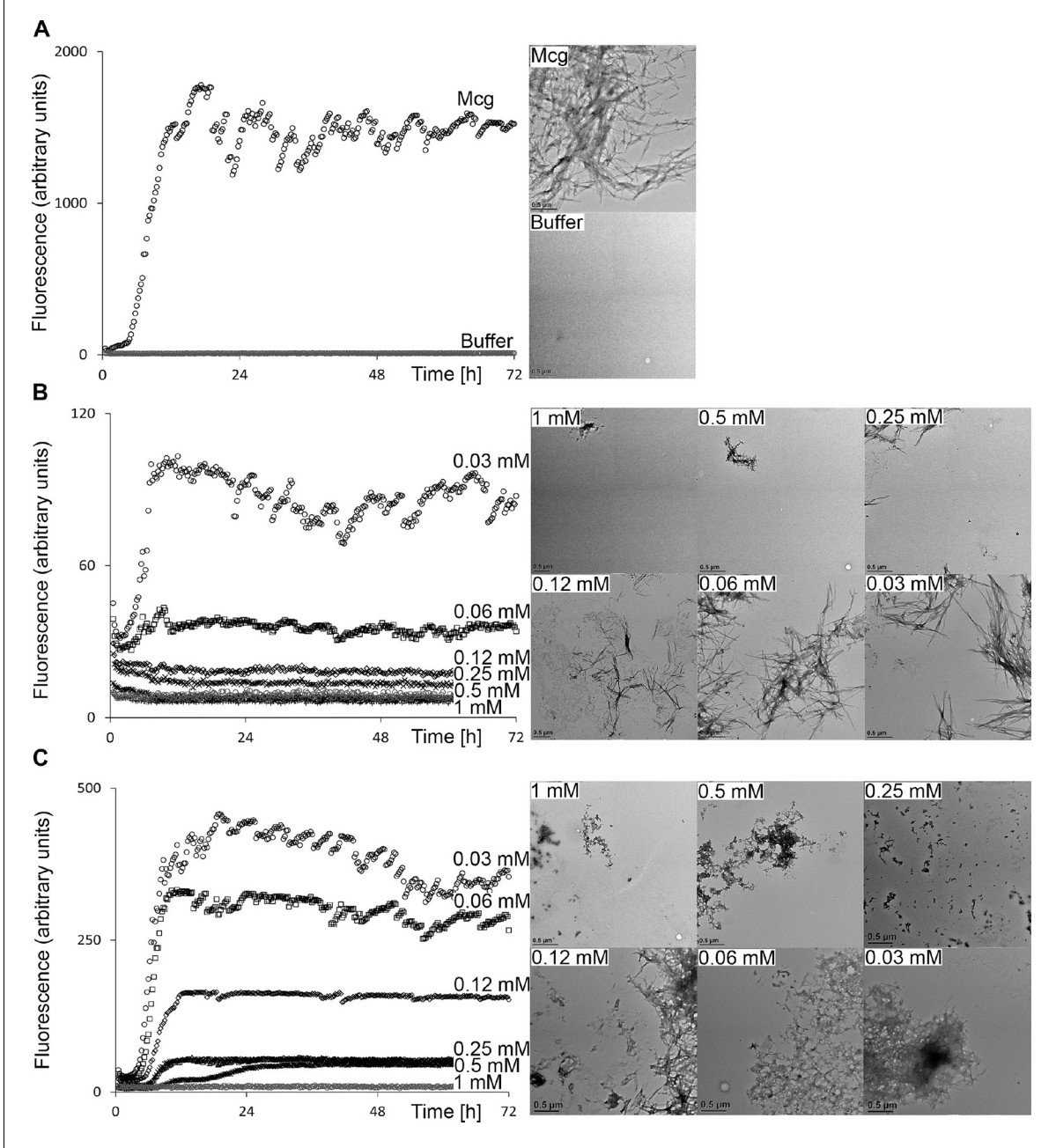

**Figure 4.** Thioflavin T fibril formation assays and electron micrographs of Mcg $V_L$s in the presence of various concentrations of methylene blue and sulfasalazine. Averaged fluorescence readings are based on three repeated ThT assays. Error bars are not shown for clarity. The EM micrographs have a scale bar of 0.5 μm. (**A**) Mcg alone. (**B**) Mcg with various concentrations of methylene blue. (**C**) Mcg with various concentrations of sulfasalazine. As the concentration of ligand is increased, the ability of $V_L$s to form amyloid fibrils is hindered, as indicated by a decrease in ThT assay fluorescence and inability to detect fibrils with EM. The ligand concentration that effectively inhibited amyloid formation in these assays was 500 μM for methylene blue and 250 μM for sulfasalazine.

meniscus depletion experiment to detect possible monomers. The experiment identified no methylene blue bound to $V_L$ monomers.

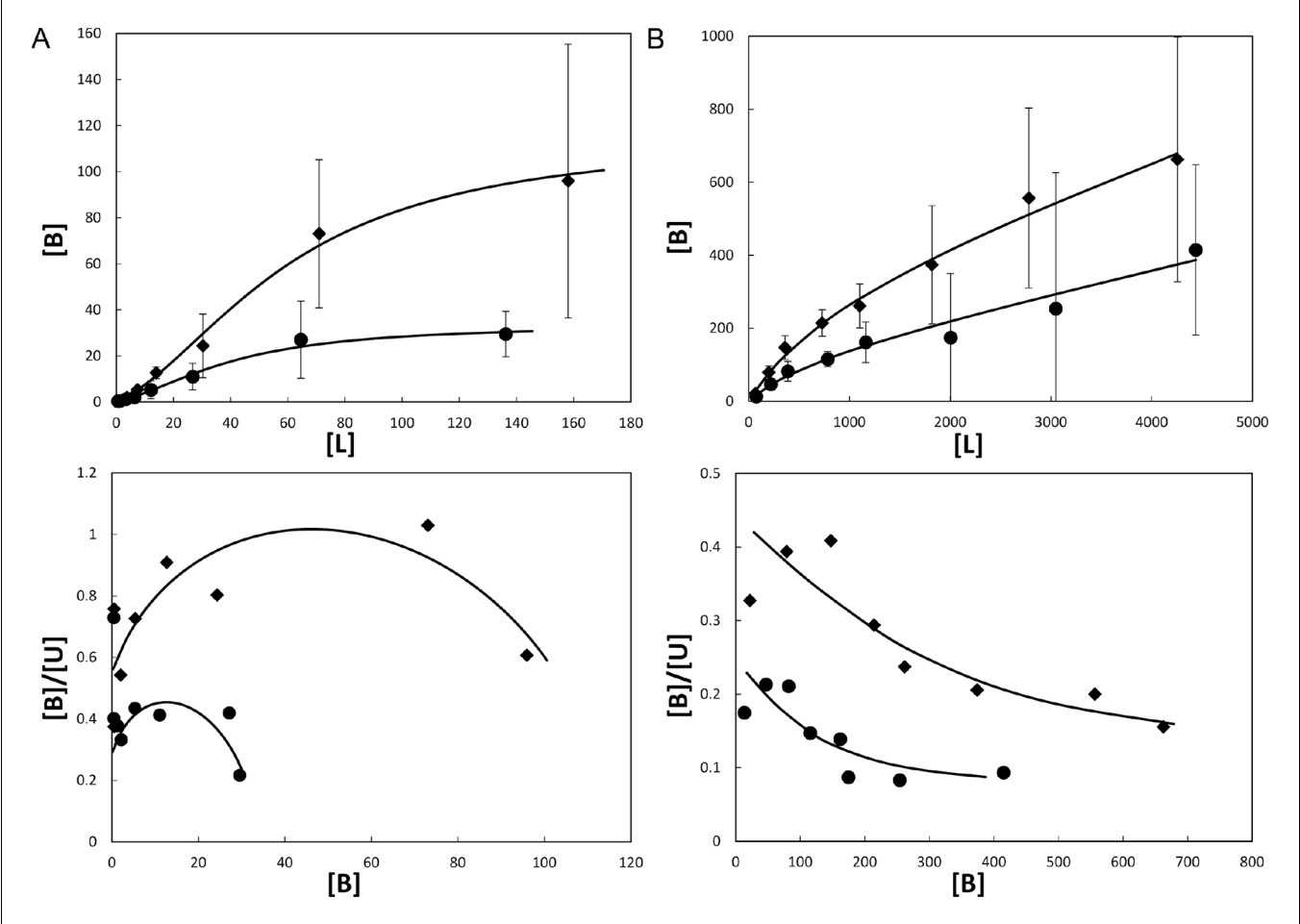

**Figure 5.** Binding of ligands to Mcg $V_L$s. Binding curves (top) and Scatchard plots (bottom) of ligand binding determined from equilibrium dialysis experiments. Each curve represents binding equations fit to the data by least squares. Binding constants were derived from the fit equations (see *Table 1*). Vertical bars represent the standard errors of the mean from independently repeated experiments. [B], [L], and [U] are bound, total, and unbound concentrations of ligand in μM. (A) Methylene blue binding to Mcg. Rhombs show means for 3 independent experiments performed with 1.0 mg/ml Mcg. Circles show means for 5 independent experiments performed with 0.5 mg/ml Mcg. Notice the sigmoidal shape of the binding plot and concave shape of the Scatchard plot indicating cooperative binding. (B) Sulfasalazine binding to Mcg. Rhombs show means for 3 independent experiments performed with 5.0 mg/ml Mcg. Circles show means for 3 independent experiments performed with 2.5 mg/ml Mcg. Notice the saturated binding and a down-curving shape of the Scatchard plot indicating one binding constant followed by non-specific binding.

## Discussion

The identification of ligands that inhibit LC amyloid formation by stabilizing the $V_L$ dimer has therapeutic implications for systemic light chain amyloidosis. By utilizing the known ability of ligands to bind the hydrophobic cavity of $V_L$ dimers, we prevent dimer disassociation into monomers and thus hinder amyloid formation (*Edmundson et al., 1984*; *1993*). A similar approach was used to design therapies for transthyretin-related amyloidosis by stabilizing the tetrameric state of the protein (*Miroy et al., 1996*). Initial screening for ligands that hinder $V_L$ amyloid formation revealed four ligands showing this effect, with methylene blue as the strongest binding ligand. A subsequent search in the DrugBank aimed to identify similarly effective ligands that are already approved for medical use.

Since the emission spectra of ThT partly overlaps with the absorption spectra of the ligands, both ThT assays and EM analysis were necessary to confirm the ability of the ligands to inhibit amyloid formation. Analysis of electron micrographs and ThT assays indicates that increasing the concentration of methylene blue or sulfasalazine hinders the formation of amyloid fibrils. One of the molecules

**Table 1.** Disassociation constants of ligands that bind to the Mcg $V_L$ dimer.

| Ligand | $\langle K_1 \rangle$ [μM] (SEM) | $\langle K_2 \rangle$ or $K_{NS}$ [μM] (SEM) | N | Binding model | Equations fit ([Y]: bound ligand, [L]: free ligand, [R]: receptor) |
|---|---|---|---|---|---|
| methylene blue | 207 (35) | 21 (4) | 8 | Two positively cooperative sites $R+L \Leftrightarrow RL+L \Leftrightarrow RL_2$ | (1) $[Y]=([L]*[R]/K_1)*(1+[L]/K_2))$ (2) $[R]=[R_0]/(1+[L]/K_1+[L]^2/(K_1*K_2))$ |
| sulfasalazine | 698 (105) | 21 (8) (non-specific binding) | 6 | Two equivalent + non-specific sites $R+L \Leftrightarrow RL$ | (3) $[Y]=[R_0]*[L]/(K_1+[L]) + K_{NS}*[L]$ |

SEM – standard error of the mean; N – number of independent experiments
$K_{NS}$ – nonspecific binding

from our screen, epigallocatechin gallate, was recently suggested to hinder the formation of amyloid fibrils, yet in our experiments it did not show any effect (*Pelaez-Aguilar et al., 2015*).

Though both of the ligands bind Mcg and induce amyloid hindering effects, the mechanism of binding slightly differs. Methylene blue binds through positive cooperativity. This result indicates the presence of two binding sites, one strong and one weak, where binding to the second site begins only after the first site is fully occupied. Scatchard plots for sulfasalazine show that the two specific binding sites on Mcg are identical with the same disassociation constant and that this specific ligand binding is followed by non-specific binding.

We addressed the difference in the binding mechanisms of methylene blue and sulfasalazine to Mcg in solution by crystal structures and analytical ultracentrifugation. The crystal structure of Mcg with methylene blue shows binding at the A-site of the $V_L$ dimer cavity and coordination by the side chains of residues Y34, Y93, D97 and F99. The crystal belongs to space group $P2_1$ with one dimer per asymmetric unit, and the symmetry between the two $V_L$ domains of Mcg is broken by the presence of a sulfate ion bound to only one of the $V_L$ domains (*Figure 6A*). Analytical ultracentrifugation analysis of the soluble quaternary state indicates the dimer as the most prevalent form of the complex versus monomer without a bound ligand (*Brumshtein et al., 2014*). We also observed a small fraction of oligomers with molecular weights larger than dimers, as confirmed by the meniscus depletion experiment. The meniscus depletion distributes quaternary states along the path length of the sample cell during analytical centrifugation, thus verifying which state accounts for the deviation in apparent

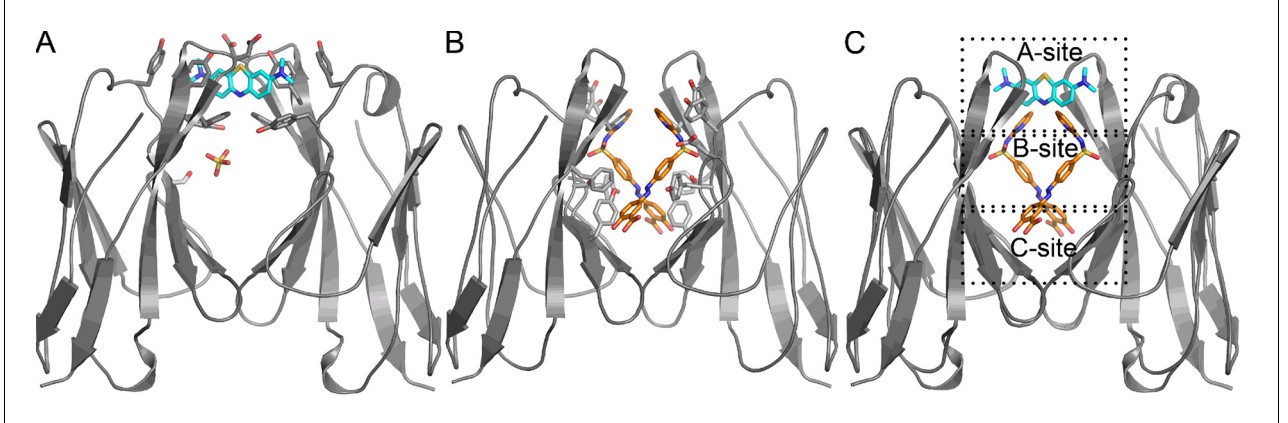

**Figure 6.** Crystal structures of Mcg binding methylene blue (PDB 5ACM) and Mcg binding sulfasalazine (PDB 5ACL). (**A**) Mcg and methylene blue. Methylene blue binds the cavity between the $V_L$ domains. The sulfate ion binds in the hydrophobic cavity at S36 and may block the secondary methylene blue binding site. (**B**) Mcg and sulfasalazine. Two symmetry-related sulfasalazine molecules bind in the hydrophobic cavity between the $V_L$ domains, simultaneously occupying all three sites: A, B, and C. (**C**) Overlay of the structures from A and B for comparison and annotation of A-B-C binding sites.

**Table 2.** Statistics of X-ray data collection and atomic refinement. Values in parentheses are for the outermost shell of data. Each structure was derived from a single crystal.

| | Mcg-methylene blue PDB 5ACM | Mcg-sulfasalazine PDB 5ACL |
|---|---|---|
| Space group | $P2_1$ | $P2_12_12$ |
| *Cell dimensions* | | |
| a, b, c (Å) | 39.2, 31.1, 73.6 | 31.8, 74.6, 39.3 |
| α, β, γ (°) | 90, 90.1, 90 | 90, 90, 90 |
| Resolution (Å) | 39.2-1.0 (1.07) | 29.3-1.5 (1.53) |
| $R_{sym}$ (%) | 7.2 (59.1) | 5.5 (52.9) |
| $I/\sigma(I)$ | 9.8 (2.4) | 18.7 (3.7) |
| Completeness (%) | 88.6 (81.8) | 97.9 (83.2) |
| Redundancy | 3.5 (3.5) | 6.9 (5.9) |
| No. reflections | 259214 | 107913 |
| $R_{work}/R_{free}$ (%) | 11.1/12.2 | 15.9/18.7 |
| *No. atoms* | | |
| Protein | 1610 | 845 |
| Ligand/ion | 36 | 33 |
| Water | 210 | 134 |
| Average B-factors (Å$^2$) | 16.8 | 20.4 |
| *R.m.s deviations* | | |
| Bond lengths (Å) | 0.05 | 0.02 |
| Bond angles (°) | 3.27 | 2.04 |
| Crystallization conditions | 0.2 M $NH_4Cl$, 2.2 M $(NH_4)_2SO_4$, 0.5 M methylene blue | 0.2 M Tri-potassium citrate, 2.2 M $(NH_4)_2SO_4$, 0.5 M sulfasalazine |

molecular weight. The most significant outcome is the complete absence of soluble monomer in complex with the ligand, which verifies our conjecture that the ligands bind and stabilize dimers. In the case of sulfasalazine, to confirm binding specifically to soluble dimers, the analytical ultracentrifugation experiments identified dimers as the only quaternary state of the ligand-bound complex.

The reconciliation of our three experiments on the binding of methylene blue to Mcg $V_L$s suggests blocking of the secondary binding site in the crystals. The three experimental findings are: cooperative binding of methylene blue as measured by equilibrium dialysis, binding of the ligand to the A-site in the crystal structure, and the absence of monomeric $V_L$ in complex with ligand. Taken together, these observations indicate that the secondary binding site of methylene blue may not be available in the crystalline state, but available in solution. Secondary binding to either the B or C-sites may require conformational changes that are less favorable in crystals or induce the higher-order assembly of complexes larger than dimers.

The crystal structure of Mcg with sulfasalazine explains why only one disassociation constant for specific binding was measured. Crystals of the complex belong to space group $P2_12_12$ with two symmetry-related ligands bound per Mcg dimer, and thus one ligand bound per $V_L$ in the asymmetric unit (*Figure 6B*). Therefore, the equivalence of the two sulfasalazine ligands in the crystal explains why only one disassociation constant was measured by means of equilibrium dialysis. The crystal structure shows that each of the sulfasalazine ligands occupy all three A-B-C-sites simultaneously; each ligand binds to the A and B-sites on one of the $V_L$s composing the dimer, while C-site binding is on the symmetry-related $V_L$. This contrasts with methylene blue, which is bound only to the A-site. Despite differences in the exact orientation of the side chains where the ligands bind, both ligands bind in the cavity of the Mcg dimer (*Figure 6C*).

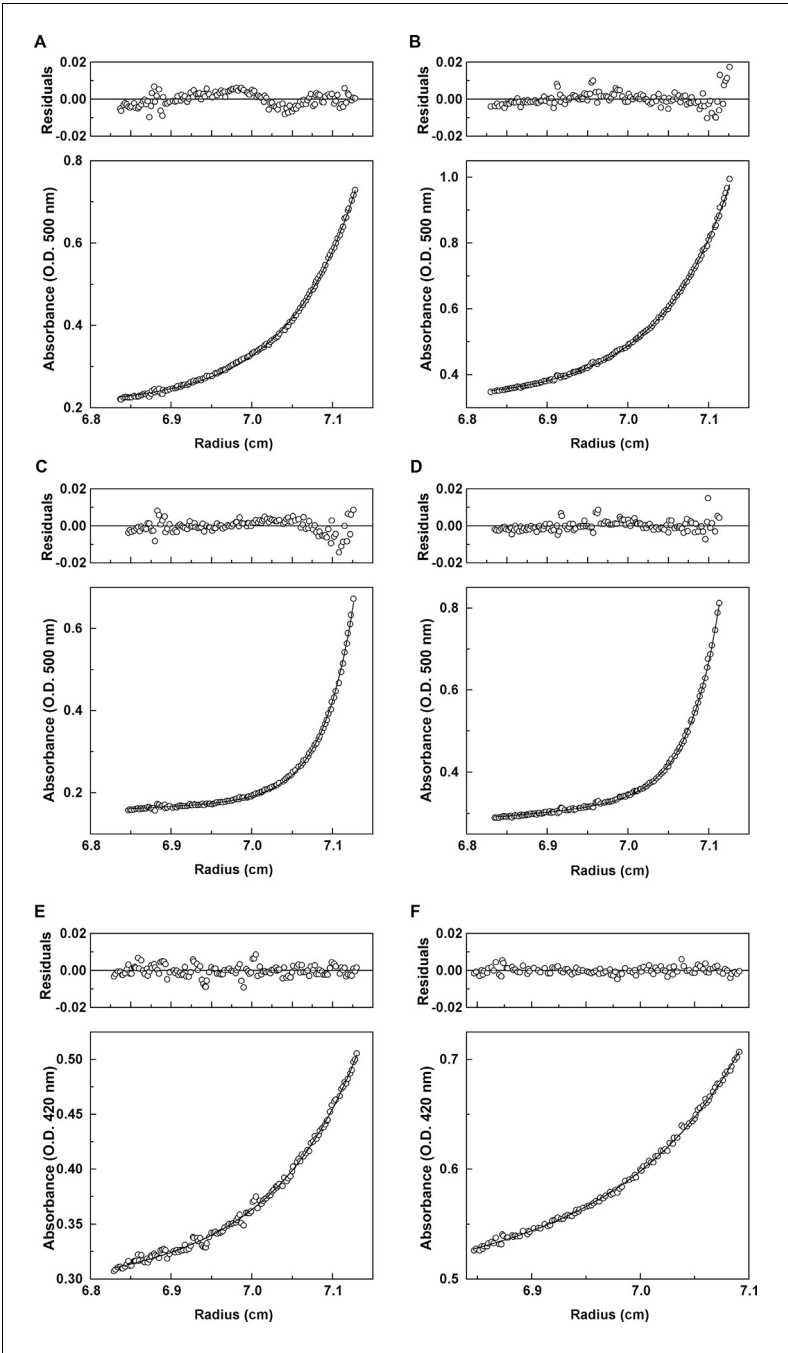

**Figure 7.** Analytical ultracentrifugation data for Mcg with methylene blue and Mcg with sulfasalazine. (**A**) Absorbance and residuals for 80 µM Mcg with 100 µM methylene blue at 22,000 rpm. (**B**) Absorbance and residuals for 80 µM Mcg with 150 µM methylene blue at 22,000 rpm. (**C**) Dimer meniscus depletion - absorbance and residuals for 80 µM Mcg with 100 µM methylene blue at 35,000 rpm. (**D**) Dimer meniscus depletion - absorbance and residuals for 80 µM Mcg with 150 µM methylene blue at 35,000 rpm. (**E**) Absorbance and residuals for 200 µM Mcg with 300 µM sulfasalazine at 22,000 rpm. (**F**) Absorbance and residuals for 200 µM Mcg with 500 µM sulfasalazine at 22,000 rpm. Measurements were taken after samples reached equilibrium and the distribution of protein along the cells did not change. Residuals show the corresponding error of the fit of the model to the data.

Although our experiments were performed using only $V_L$s rather than full-length LCs, the ligand-binding site between $V_L$s is maintained in full-length LCs. While full-length LCs are disulfide linked at

**Table 3.** Apparent $V_L$ molecular weights and ligand binding derived from analytical ultracentrifugation.

| Speed (rpm) | Concentration of Mcg (μM) | Ligand | Concentration of ligand (μM) | Apparent molecular weight (Da) | Oligomer state |
|---|---|---|---|---|---|
| 22,000 | 80 | methylene blue | 100 | 27,342 ± 268* | Dimer + multimer |
| | | | 150 | 26,507 ± 225* | Dimer + multimer |
| 35,000 | 80 | methylene blue | 100 | 23,609 ± 212* | Dimer |
| | | | 150 | 23,397 ± 152* | Dimer |
| 22,000 | 200 | sulfasalazine | 300 | 21,989 ± 571* | Dimer |
| | | | 500 | 21,821 ± 345* | Dimer |

*Standard error.

their $C_L$ C-termini, crystal structures indicate that the conformation of $V_L$s is independent of connected constant domains (*Ely et al., 1989*; *Terzyan et al., 2003*; *Makino et al., 2007*). In full-length LCs, the joining (J) segment between $V_L$s and $C_L$s provides the flexibility for $V_L$s to maintain the same hydrophobic cavity as that seen in the $V_L$ homo-dimer alone. This indicates the strategy of using ligands to stabilize the $V_L$ dimer and hinder amyloid fibril formation may also be effective for full-length homo-dimers.

The ability of ligands to hinder the aggregation of $V_L$s into amyloid fibrils by stabilizing $V_L$ dimers indicates a promising approach for treatment of systemic light chain amyloidosis disease. While the light chain amino acid sequence differs between patients, the characteristic hydrophobic cavity of the homo-dimer formed by overexpressed LCs is conserved (*Bodi et al., 2009*; *Bork et al., 1994*; *Edmundson et al., 1976*; *Solomon and Weiss, 1995*). Although our experiments were performed using a single $V_L$ variant, LCs of both λ and κ maintain a similar tertiary structure and hydrophobic cavity (*Qin et al., 2007*; *Wetzel, 1994*; *Brumshtein et al., 2014*; *Bellotti et al., 2000*; *Fink, 1998*; *Khurana et al., 2001*). This conserved hydrophobic pocket indicates a possible target for designing medications that inhibit LC amyloidosis. Although methylene blue and sulfasalazine may not bind tightly enough to be directly repurposed as treatments for the disease, these ligands may serve as prototypes for development of effective medications. As indicated by the ability of these molecules to inhibit amyloid fibril formation, this approach may result in alleviated symptoms for systemic light chain amyloidosis patients.

## Materials and methods

### Preparation of recombinant proteins

Protein samples were prepared as previously described (*Brumshtein et al., 2014*). In summary, the $V_L$ of the pathologic LC homo-dimer Mcg was expressed in *E. Coli*, purified in denaturing conditions, refolded, and concentrated.

### Identification of ligands that impede amyloid fibril formation

Following previous research demonstrating the ability of ligands to bind in the hydrophobic cavity between the two $V_L$ domains of the dimer, aromatic and other hydrophobic ligands were screened for an inhibitory effect on the formation of amyloid fibrils (*Edmundson et al., 1984*;*1993*). Amyloid formation was monitored with Thioflavin T assays and electron microscopy analysis, and we used molecules that inhibited amyloid fibril formation at concentrations below 1 mM to search DrugBank to identify biomedical compounds with a similar structure that would potentially inhibit the formation of amyloid fibrils (*Law et al., 2014*). We selected structures in DrugBank with a similarity threshold of 0.3. Search filters were not applied to any other parameters. We dissolved the identified molecules in either water or 20% DMSO for subsequent use in assays of amyloid formation.

## Amyloid fibril formation assays

Thioflavin (ThT) assays of Mcg amyloid formation were performed in acidic conditions as previously described (*Bellotti et al., 2000*). Assays were conducted in the presence of varying concentrations of the ligands (1.0, 0.50, 0.25, 0.12, 0.05, and 0.03 mM) at 37°C and constant 300 rpm shaking with Teflon balls with 0.125 inch radii as stirrers. ThT fluorescence spectra were recorded at 440/480 nm excitation/emission wavelengths (*Wall et al., 1999*).

## Electron microscopy (EM)

Samples from the ThT fibril formation assays were diluted with water to 10% v/v and applied onto copper grids with formvar-carbon coating (Ted Pella, Inc., Redding, CA, Cat. No. 01810). Negative staining was performed with 2% w/v uranyl acetate and images were collected by means of a Tecnai T12 electron microscope at 120 kV with a Gatan CCD camera.

## Equilibrium dialysis binding

Binding constants of the ligands with Mcg were derived using Rapid Equilibrium Dialysis plates with a membrane cutoff of 8 kDa (Thermo Scientific Inc., Cat. No. 90006) (*Waters et al., 2008*). Methylene blue at concentrations ranging from ~1–200 μM with 0.5 and 1.0 mg/ml of Mcg, and sulfasalazine at concentrations ranging from ~100–5000 μM with 2.5 and 5.0 mg/ml of Mcg were incubated for 72 hours with constant agitation in standard laboratory conditions. Following incubation, samples were collected from the membrane and reference cells and final concentrations were measured by spectrophotometry.

## Crystal structure determination

Co-crystallization trials were set up with a Mosquito micro-crystallization robot, using a solution of 20 mg/ml Mcg with 1 mM methylene blue or sulfasalazine ligands. All crystals appeared in the presence of sulfate or phosphate anions. Crystals were cryo-protected with 25% glycerol and X-ray data were collected at the 24-ID-C beamline of the Advanced Photon Source (Mcg-methylene blue) or with a Rigaku Raxis4$^{++}$ with a HCT detector (Mcg-sulfasalazine). X-ray diffraction data were processed with XDS package (*Kabsch, 2010*). Initial phases were obtained by means of molecular replacement using PDB 3MCG as the initial model and structures were refined with Refmac5 (*Murshudov et al., 1997*). Graphics were rendered with PyMol and refined atomic models were deposited into the Protein Data Bank (PDB) (*DeLano, 2002*).

## Analytical ultracentrifugation

Sedimentation equilibrium experiments were performed on Mcg in the presence of methylene blue and sulfasalazine. Equilibrium runs were performed at 20°C in a Beckman Optima XL-A analytical ultracentrifuge using absorption optics at 500 nm for methylene blue and 420 nm for sulfasalazine. Samples were at concentrations of 1 mg/ml of Mcg with 100 and 150 μM of methylene blue, and 1 and 2.5 mg/ml of Mcg with 300 and 500 μM sulfasalazine. The concentrations of the protein and ligands were chosen to obtain optical density readings in the linear range of the spectrophotometer yet with measurable quaternary state densities of the complexes.

Sedimentation equilibrium profiles were measured at speeds of 10,000, 14,000 and 22,000 rpm in a An-60 Ti rotor. To validate the deviation of the apparent molecular weight of Mcg with methylene blue, a set of experiments analyzing the meniscus depletion of the Mcg dimer were performed at 35,000 rpm. The data were fit with a non-linear least squares double exponential fit using the Beckman Origin-based software (Version 3.01) (*Cohn and Edsall, 1943*; *Laue et al., 1992*).

## Acknowledgements

We thank Mike Collazo for crystallization screens at the UCLA-DOE crystallization facility, M Capel, K Rajashankar, N Sukumar, J Schuermann, I Kourinov, and F Murphy for facilitating X-ray data collection experiments at the NE-CAT beamline 24-ID-C of APS, and Dr. Julian Whitelegge and Dr. Meytal Landau for discussions. We thank HHMI, NIH AG 04812 and NIGMS R25GM055052 for support. We thank Dr. Jeffery W Kelly and other eLife reviewers for their comments that helped us to improve this manuscript.

## Additional information

### Funding

| Funder | Grant reference number | Author |
|---|---|---|
| National Institutes of Health | AG048120 | Boris Brumshtein<br>Shannon R Esswein<br>Lukasz Salwinski<br>Martin L Phillips<br>Alan T Ly<br>Duilio Cascio<br>Michael R Sawaya<br>David S Eisenberg |
| National Institute of General Medical Sciences | R25GM055052 | Alan T Ly |

The funders had no role in study design, data collection and interpretation, or the decision to submit the work for publication.

### Author contributions

BB, SRE, Conception and design, Acquisition of data, Analysis and interpretation of data, Drafting or revising the article, Contributed unpublished essential data or reagents; LS, DSE, Conception and design, Analysis and interpretation of data, Drafting or revising the article; MLP, DC, MRS, Acquisition of data, Analysis and interpretation of data, Drafting or revising the article, Contributed unpublished essential data or reagents; ATL, Acquisition of data, Drafting or revising the article, Contributed unpublished essential data or reagents

### Author ORCIDs

Shannon R Esswein, http://orcid.org/0000-0002-5142-0190

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
