## [Decision Letter]

Thank you for submitting your work entitled "Inhibition by small-molecule ligands of formation of amyloid fibrils of variable domains of IgG light chains" for peer review at *eLife*. Your submission has been favorably evaluated by Randy Schekman (Senior editor) and two reviewers, one of whom (Jeffery W. Kelly) is a member of our Board of Reviewing Editors.

The Reviewing editor has drafted this decision letter to help you prepare a revised submission.

Both the reviewer and the Reviewing editor agree that the authors have demonstrated an important proof-of-principle regarding the potential utility of a dimer stabilizing therapeutic approach for light chain amyloidosis, which has long being theorized for systemic AL amyloidosis (see Bellotti et al., J Struct Biol 2000).

We seek a minor revision from the authors: addressing all the questions/suggestions posed in the two reviews and two minor comments below.

Reviewer #1:

The manuscript by Brumshtein et al. entitled "Inhibition by small-molecule ligands of formation of amyloid fibrils of variable domains of Ig light chains" is an excellent paper that merits serious consideration for publication in *eLife*. The authors have accomplished a feat that many groups have tried to accomplish previously – i.e. to stabilize the light chain variable domain dimer which is non-amyloidogenic, inhibiting dimer dissociation to an amyloidogenic monomer.

While the light chain dimer sequence varies between patients, the hydrophobic cavity at the variable domain dimer-dimer interface is conserved in sequence. The light chain variable domain dimer dissociates, misfolds and aggregates driving degenerative disease phenotypes. Thus, the authors screened a library of small molecules previously implicated to bind at the variable domain dimer interface. Analysis of electron micrographs and thioflavin T assays reveals that increasing the concentration of methylene blue or sulfasalazine hinders the formation of light chain aggregation. Two ligands bind to each dimer, methylene blue binding with positive cooperativity, whereas sulfasalazine binds non-cooperatively. Analytical ultracentrifugation is consistent with ligand binding to and stabilization of the dimer. In the crystal structure only one molecule of methylene blue binds / dimer vs. two molecules of sulfasalazine / dimer. While these two ligands do not bind tightly enough to become drug candidates, they convincingly demonstrate proof-of-principle that molecules that bind to the non-amyloidogenic variable domain light chain dimer prevent light chain aggregation driving light chain amyloidosis. This data provides the incentive for a commercial organization to make a non-amyloidogenic light chain dimer stabilizer.

Can the authors please comment on whether this strategy could be useful for inhibition of the amyloidogenesis from the disulfide-linked full-length light chain dimer, wherein each chain comprises 1 constant and 1 variable domain, i.e., is the same binding site conserved?

Reviewer #1 (Minor Comments):

1) The first sentence of the Introduction is a run-on sentence that should be rewritten as two sentences: "Amyloid fibrils are protein deposits, which are often associated with disease and show common structural and biochemical characteristics, including the aggregation into proteins into non-covalent self-complementary fibrils, resistance to proteolysis, insolubility in aqueous solution, binding of thioflavin dye, and generation of a cross-beta X-ray diffraction pattern."

2) That dimer dissociation and partial monomer denaturation drives variable domain aggregation was shown by Wetzel et al. (early 80's) and this literature should be referenced.

3) In Figure 2 it would be best to depict the dissociated amyloidogenic monomers as having a shape other than that in the non-amyloidogenic dimer.

*Reviewer #2:*

Brumshtein and colleagues report the identification of ligands (including approved biomedical compounds) that bind and stabilize dimers of a human amyloidogenic immunoglobulin λ light chain (Mcg), thereby inhibiting amyloid formation in vitro.

This is an important proof-of-principle study about the potential utility of such a therapeutic approach, which is currently being pursued for ATTR amyloidosis and which has long being theorized also for systemic AL amyloidosis (see Bellotti et al., J Struct Biol 2000).

As discussed by the authors, each patient with systemic AL amyloidosis has a unique immunoglobulin light chain sequence. However, the study is focused on an individual amyloidogenic immunoglobulin λ light chain (Mcg). Whereas this is acceptable for a proof-of-principle study (even though analysis of other amyloidogenic light chains, including also κ light chains, would represent a significant improvement) this fact should be better discussed in the manuscript, and should also be reflected in the title and in the Abstract.

Reviewer #2 (Minor Comments):

1) Title: "Inhibition by small-molecule ligands of formation of amyloid fibrils of variable domains of IgG light chains". The abbreviation IgG (immunoglobulin of G type) is inappropriate in this case and should be replaced with immunoglobulin. To take into account the above mentioned issue of having investigated only Mcg, a new suggested title could be: "Inhibition by small-molecule ligands of formation of amyloid fibrils of variable domains of Mcg immunoglobulin light chains". This would also be in line with the work done by the group of Edmundson on Mcg, where the focus on Mcg is reflected in the titles of the different papers.

2) Introduction: "[…] including the aggregation into proteins into non covalent self complementary fibrils […]". This is not clear, and should be rephrased.

3) Introduction: "Systemic light chain amyloidosis, frequently associated with multiple myeloma, was first described in 1848 when Dr. Bence Jones discovered a disease associated substance in urine […]". Systemic AL amyloidosis is only occasionally associated with multiple myeloma and was not discovered first by Dr. Bence Jones (see Kyle, Amyloid 2011).

4) Introduction: "This disease is characterized by overexpression of monoclonal LCs, which are able to form homo dimers termed Bence Jones proteins […]". The use of Bence Jones protein referred to homo dimers of monoclonal light chain is inappropriate.

5) Introduction: "LCs are produced by white blood cells […]" is too generic. Use Plasma cells instead of white blood cells.

6) Introduction: "The mechanism of amyloid formation also suggests that shifting the equilibrium away from the amyloid prone monomer by stabilizing the dimer would hinder formation of amyloid fibrils (Figure 2). (Bulawa et al., 2012)." Quote also Bellotti et al., J Struct Biol 2000.

7) Introduction: "The monomer dimer equilibrium of VLs suggests that systemic LC amyloidosis […]". The use of systemic AL amyloidosis, according to international guidelines on nomenclature, would be preferable (see Sipe et al., Amyloid 2014).

8) Materials and methods: The addition of a short paragraph on the in silico search performed in the Drugbank website, including which type of search (ChemQuery? Other?), search options and applied filters, would be useful.

9) Figure 4: Replace 0.5 μM with 0.5 μm in the scale bar description. Define FI in the legend. Add units of measure in all y axes of panels A, B and C.

10) Figure 5: Define [B], [L] and [B]/[U] in the legend. Add units of measure in all x and y axes of panels A and B.

11) Figure 7: Add units of measure in all y axes of panels A to F.

12) Discussion: "The most significant outcome is the complete absence of soluble monomer in complex with the ligand, which verifies our conjecture that the binding of ligands to VLs induces dimerization". Does binding of ligands to VLs induce dimerization or do ligands bind and stabilize dimers?

13) Discussion: "While the light chain amino acid sequence differs between patients, the characteristic hydrophobic cavity of the VL homo dimer formed by overexpressed LCs is conserved". This point is central to the more broaden applicability of the therapeutic approach suggested in this study. This point should be better explained and appropriate references should be quoted to circumstantiate this claim.

---

## [Author Response]

Reviewer #1:

[…] Can the authors please comment on whether this strategy could be useful for inhibition of the amyloidogenesis from the disulfide-linked full-length light chain dimer, wherein each chain comprises 1 constant and 1 variable domain, i.e., is the same binding site conserved?

We added a paragraph explaining the topic before the last paragraph of the Discussion. The modified text is: “Although our experiments were performed using only VLs rather than full-length LCs, the ligand-binding site between VLs is maintained in full-length LCs. […] This indicates the strategy of using ligands to stabilize the VL dimer and hinder amyloid fibril formation may also be effective for full-length homo-dimers.”

Reviewer #1 (Minor Comments):

*1) The first sentence of the Introduction is a run-on sentence that should be rewritten as two sentences: "Amyloid fibrils are protein deposits, which are often associated with disease and show common structural and biochemical characteristics, including the aggregation into proteins into non-covalent self-complementary fibrils, resistance to proteolysis, insolubility in aqueous solution, binding of thioflavin dye, and generation of a cross-beta X-ray diffraction pattern."*

We split the run-on sentence into two and polished them. The modified text is: “Amyloid fibrils are protein deposits often associated with disease. These deposits show common structural and biochemical characteristics, including non-covalent self-complementary fibrils, resistance to proteolysis, insolubility in aqueous solution, binding of thioflavin dye, and generation of a cross-β X-ray diffraction pattern.”

*2) That dimer dissociation and partial monomer denaturation drives variable domain aggregation was shown by Wetzel et al. (early 80's) and this literature should be referenced.*

We referenced Wetzel et al.

*3) In Figure 2 it would be best to depict the dissociated amyloidogenic monomers as having a shape other than that in the non-amyloidogenic dimer.*

We prefer to keep the original figure, since monomers are in equilibrium with dimers in solution and the tertiary structure is maintained unless the monomer proceeds to unfold and form amyloid fibers.

Reviewer #2:

*[…] As discussed by the authors, each patient with systemic AL amyloidosis has a unique immunoglobulin light chain sequence. However, the study is focused on an individual amyloidogenic immunoglobulin λ light chain (Mcg). Whereas this is acceptable for a proof-of-principle study (even though analysis of other amyloidogenic light chains, including also κ light chains, would represent a significant improvement) this fact should be better discussed in the manuscript, and should also be reflected in the title and in the Abstract.*

We modified the title to: “Inhibition by small-molecule ligands of formation of amyloid fibrils of an immunoglobulin light chain variable domains”

We modified the last sentence of the Abstract to: “Here we identify ligands that inhibit amyloid formation by stabilizing the Mcg light chain variable domain dimer and shifting the equilibrium away from the amyloid-prone monomer.”

We added an explanation to the last paragraph of the Discussion. The new text is: “The ability of ligands to hinder the aggregation of VLs into amyloid fibrils by stabilizing VL dimers indicates a promising approach for treatment of systemic light chain amyloidosis disease. […] As indicated by the ability of these molecules to inhibit amyloid fibril formation, this approach may result in alleviated symptoms for systemic light chain amyloidosis patients.”

Reviewer #2 (Minor Comments):

*1) Title: "Inhibition by small-molecule ligands of formation of amyloid fibrils of variable domains of IgG light chains". The abbreviation IgG (immunoglobulin of G type) is inappropriate in this case and should be replaced with immunoglobulin. To take into account the above mentioned issue of having investigated only Mcg, a new suggested title could be: "Inhibition by small-molecule ligands of formation of amyloid fibrils of variable domains of Mcg immunoglobulin light chains". This would also be in line with all the work done by the group of Edmundson on Mcg, where the focus on Mcg is reflected in the titles of the different papers.*

See above.

2) Introduction: "[...] including the aggregation into proteins into non covalent self complementary fibrils [...]". This is not clear, and should be rephrased.

We reworded the sentence for clarity. The new text is: “These deposits show common structural and biochemical characteristics, including non-covalent self-complementary fibrils, resistance to proteolysis, insolubility in aqueous solution, binding of thioflavin dye, and generation of a cross-β X-ray diffraction pattern.”

3) Introduction: "Systemic light chain amyloidosis, frequently associated with multiple myeloma, was first described in 1848 when Dr. Bence Jones discovered a disease associated substance in urine [...]". Systemic AL amyloidosis is only occasionally associated with multiple myeloma and was not discovered first by Dr. Bence Jones (see Kyle. Amyloid 2011).

We rewrote the sentence and used the nomenclature recommended by Sipe et al., and added references to Kyle. Amyloid 2011 and Sipe et al. Amyloid 2014. The new text is: “Cases of systemic light chain amyloidosis (AL), associated with multiple myeloma, were first described during the Renaissance though the etiology of the disease was not clear. In 1848, Dr. Bence-Jones discovered a disease-associated substance in urine, later identified as immunoglobulin light chains (LC) (Kyle, 2001; Kyle, 2011; Dahlin and Dockerty, 1950; Glenner et al., 1970; Glenner et al., 1971; Glenner, 1973; Sipe et al., 2014).”

We added the reference to Kyle as a historical overview and kept the reference to Bence-Jones as he identified the substance in urine as protein.

To clarify that light-chain amyloidosis is only occasionally associated with multiple myeloma, we removed the word “frequently.” Also, we polished the paragraph due to the changes and the modified sentences are: “Cases of systemic light chain amyloidosis (AL), associated with multiple myeloma, were first described in the mid-19th century; however, the etiology of the disease was not clear. In 1848, Dr. Bence-Jones discovered the disease-associated substance in urine, though it was only later identified as immunoglobulin light chains (LC) associated with amyloidosis (Kyle, 2001; Kyle, 2011; Dahlin and Dockerty, 1950; Glenner et al., 1970; Glenner et al., 1971; Glenner, 1973; Sipe et al., 2014).”

*4) Introduction: "This disease is characterized by overexpression of monoclonal LCs, which are able to form homo dimers termed Bence Jones proteins [...]". The use of Bence Jones protein referred to homo dimers of monoclonal light chain is inappropriate.*

We removed the term Bence-Jones proteins. The new text is: “Systemic AL amyloidosis is characterized by overexpression of monoclonal LCs, which are able to form homo-dimers.”

5) Introduction: "LCs are produced by white blood cells [...]" is too generic. Use Plasma cells instead of white blood cells.

We accepted the reviewer’s suggestion. The text now reads: “LCs are produced by plasma cells and their final amino acid sequence is determined by somatic recombination (Sakano et al., 1979 and Marchalonis and Schluter, 1989).”

6) Introduction: "The mechanism of amyloid formation also suggests that shifting the equilibrium away from the amyloid prone monomer by stabilizing the dimer would hinder formation of amyloid fibrils (Figure 2). (Bulawa et al., 2012)." Quote also Bellotti et al., J Struct Biol 2000.

Belloti et al. J Struc Biol 2000 reference added.

*7) Introduction: "The monomer dimer equilibrium of VLs suggests that systemic LC amyloidosis [...]". The use of systemic AL amyloidosis, according to international guidelines on nomenclature, would be preferable (see Sipe et al., Amyloid 2014).*

We changed the nomenclature to that recommended by Sipe et al., Amyloid 2014. The reference was added earlier in the Introduction. New text is: “The monomer-dimer equilibrium of VLs suggests that systemic AL amyloidosis may be mitigated by binding ligands to the cavity at the VL dimer interface (Figure 2).”

*8) Materials and methods: The addition of a short paragraph on the in silico search performed in the Drugbank website, including which type of search (ChemQuery? Other?), searching options and applied filters, would be useful.*

We expanded our explanation of how we used the DrugBank service. Since we did not want to apply too many filters on potential results, we only used a single search parameter. The new text is: “Amyloid formation was monitored with Thioflavin T assays and electron microscopy analysis, and we used molecules that inhibited amyloid fibril formation at concentrations below 1 mM to search DrugBank to identify biomedical compounds with a similar structure that would potentially inhibit the formation of amyloid fibrils (Edmundson et al., 1976). We selected structures in DrugBank with a similarity threshold of 0.3. Search filters were not applied to any other parameters. We dissolved the identified molecules in either water or 20% DMSO for subsequent use in assays of amyloid formation.”

*9) Figure 4: Replace 0.5*
μ*M with 0.5*
μ*m in the scale bar description. Define FI in the legend. Add units of measure in all y axes of panels A, B and C.*

We replaced the units in the legend, defined Fl as fluorescence, and added units on y-axes on the graph.

*10) Figure 5: Define [B], [L] and [B]/[U] in the legend. Add units of measure in all x and y axes of panels A and B.*

We added units of measure in the legend. New sentence is: “[B], [L], and [U] are bound, total, and unbound concentrations of ligand in μM.”

*11) Figure 7: Add units of measure in all y axes of panels A to F.*

Added units at y axes.

*12) Discussion: "The most significant outcome is the complete absence of soluble monomer in complex with the ligand, which verifies our conjecture that the binding of ligands to VLs induces dimerization". Does binding of ligands to VLs induce dimerization or do ligands bind and stabilize dimers?*

We reworded this sentence to better describe the effect of ligand binding. The new text is: “The most significant outcome is the complete absence of soluble monomer in complex with the ligand, which verifies our conjecture that the ligands bind and stabilize dimers.”

*13) Discussion: "While the light chain amino acid sequence differs between patients, the characteristic hydrophobic cavity of the VL homo dimer formed by overexpressed LCs is conserved". This point is central to the more broaden applicability of the therapeutic approach suggested in this study. This point should be better explained and appropriate references should be quoted to circumstantiate this claim.*

We added an explanation to the last paragraph of the Discussion. The new text is: “The ability of ligands to hinder the aggregation of VLs into amyloid fibrils by stabilizing VL dimers indicates a promising approach for treatment of systemic light chain amyloidosis disease. […] Although our experiments were performed using a single VL variant, LCs of both λ and κ maintain a similar tertiary structure and hydrophobic cavity (Qin et al., 2007; Wetzel, 1994; Brumshtein et al., 2014; Bellotti, Mangione, and Merlini, 2000; Cohn and Edsall, 1943; Laue et al., 1992).”